

# BinSanity: unsupervised clustering of environmental microbial assemblies using coverage and affinity propagation

Elaina D. Graham[1], John F. Heidelberg[1] and Benjamin J. Tully[1,2]

[1] Department of Biological Sciences, University of Southern California, Los Angeles, CA, USA
[2] Center for Dark Energy Biosphere Investigations, Los Angeles, CA, USA

## ABSTRACT

Metagenomics has become an integral part of defining microbial diversity in various environments. Many ecosystems have characteristically low biomass and few cultured representatives. Linking potential metabolisms to phylogeny in environmental microorganisms is important for interpreting microbial community functions and the impacts these communities have on geochemical cycles. However, with metagenomic studies there is the computational hurdle of 'binning' contigs into phylogenetically related units or putative genomes. Binning methods have been implemented with varying approaches such as k-means clustering, Gaussian mixture models, hierarchical clustering, neural networks, and two-way clustering; however, many of these suffer from biases against low coverage/abundance organisms and closely related taxa/strains. We are introducing a new binning method, BinSanity, that utilizes the clustering algorithm affinity propagation (AP), to cluster assemblies using coverage with compositional based refinement (tetranucleotide frequency and percent GC content) to optimize bins containing multiple source organisms. This separation of composition and coverage based clustering reduces bias for closely related taxa. BinSanity was developed and tested on artificial metagenomes varying in size and complexity. Results indicate that BinSanity has a higher precision, recall, and Adjusted Rand Index compared to five commonly implemented methods. When tested on a previously published environmental metagenome, BinSanity generated high completion and low redundancy bins corresponding with the published metagenome-assembled genomes.

# INTRODUCTION

Studies in microbial ecology commonly experience a bottleneck effect due to difficulties in microbial isolation and cultivation (*Staley & Konopka, 1985*). Due to the difficulty in culturing most organisms in a laboratory setting, alternative methods to analyze microbial diversity are commonly used to elucidate community structure and putative functionality. One such method is the sequencing of the collective genomes (metagenomics) of all microorganisms in an environment (*Handelsman et al., 1998*). Metagenomics can elucidate genomic potential, providing information on pathways, metabolism, and taxonomy allowing for inferences about environmental context without cultivation (*Meyer et al.,*

Corresponding author
Elaina D. Graham,
Egraham147@gmail.com

*2016*). Grouping contigs into metagenome-assembled genomes (MAGs) is one of the hurdles faced when analyzing metagenomic data. Typically, one of a few issues are encountered in current binning protocols, including: decreasing accuracy for contigs below a size threshold, necessity of human intervention in distinguishing clusters, struggling to differentiate related microorganisms, or excluding low coverage and low abundance organisms (*Alneberg et al., 2014*; *Bowers et al., 2015*; *Imelfort et al., 2014*).

Popular unsupervised binning methods commonly use compositional parameters, such as tetranucleotide frequency (*Anantharaman, Breier & Dick, 2016*; *Pride et al., 2003*; *Tully & Heidelberg, 2016*; *Tully et al., 2014*), as the major delimiting parameter for creating putative groups of related sequences (bins). Due to the taxon specific nature of codon usage (*Chen et al., 2004*; *Kanaya et al., 1999*), GC content (*Bohlin et al., 2010*; *Chen et al., 2004*), and short oligonucleotides (k-mers) (*Sandberg et al., 2001*; *Zhou, Olman & Xu, 2008*), these fingerprints have been used to characterize and cluster contigs. However, the utilization of composition alone can lead to biases during the binning process for a number of reasons, including, closely related species having similar fingerprints and/or recently acquired genes from horizontal transfer, which can create chimeric bins that do not represent reality (*Dick et al., 2009*). Several methods and protocols have had increased success by incorporating coverage information as an additional variable during binning (*Alneberg et al., 2014*; *Imelfort et al., 2014*; *Kang et al., 2015*; *Lu et al., 2016*; *Wu et al., 2014*). Development of new binning protocols are essential for characterizing complex environmental communities and exploring microbial diversity at a level that cultivation-based studies presently cannot achieve.

BinSanity utilizes the clustering algorithm Affinity Propagation (AP) and accepts contig coverage values as the primary delimiting component. While other clustering algorithms can effectively group related DNA fragments using composition and coverage data, common methods, like hierarchical and k-means clustering, require human input of information criteria that dictate the ultimate number of clusters (e.g., Bayesian information criterion). Assigning an *a prio*ri number for community diversity is increasingly difficult in complex ecosystems. AP, in contrast, requires no input on determining cluster centers; instead every point is iteratively considered as a potential cluster center. Data shows that AP effectively clusters a variety of data types and is often more precise than similar clustering methods (*Chen-Chia et al., 2015*; *Flynn & Moneypenny, 2013*; *Frey & Dueck, 2007*; *Fujiwara et al., 2015*; *Gan & Ng, 2015*; *Hassanabadi et al., 2014*; *Leone, Sumedha & Weigt, 2007*; *Walter, Fischer & Buhmann, 2007*; *Zhengdong & Carreira-Perpinan, 2008*). Though the implementation of AP for clustering contigs has been used before (*Lin & Liao, 2016*), the primary method of clustering involved two composition based metrics, single copy marker genes and tetranucleotide frequencies. BinSanity, in contrast, bypasses possible composition based biases for binning contigs by creating an initial set of clusters determined using coverage. When necessary, these clusters can be refined with a composition based approach to deconvolute organisms with converging abundance values.

We benchmarked BinSanity by comparing it to five currently published binning software tools. We constructed several artificial microbial communities and created *in silico* metagenomic samples based on these sequences. The communities were composed of

sequences that could be problematic for composition based binning algorithms, specifically metagenomes consisting of closely related and low abundance organisms. Additionally, a dataset associated with an infant gut microbiome time-series was used to establish how clusters generated via BinSanity compared to a highly curated set of genomic bins originally constructed using emergent self-organizing maps (ESOMs) (*Dick et al., 2009*). The results of this study find that BinSanity can generate high-quality genomes from metagenomics datasets via an automated process, which will enhance our ability to understand complex microbial communities.

## METHODOLOGY

### Artificial metagenomes

In total, 60 reference genomes (including some closed genomes, some MAGs, and some draft genomes; Table S1), consisting of a variety of organisms with ecological and environmental significance, were accessed from the Joint Genome Institute (JGI) Integrated Microbial Genome (IMG) Portal (*Markowitz et al., 2014*) and NCBI (*Pruitt, Tatusova & Maglott, 2007*). These genomes were used to create *in silico* microbial communities. Reference genomes were screened via CheckM (*Parks et al., 2015*) to provide values of completion and contamination/redundancy based on single copy genes. Several combinations of the reference genomes were used to construct artificial communities (see below). For each community, *in silico* metagenomes were generated using the reads-for-assembly script (https://github.com/meren/reads-for-assembly), which generates "Illumina-like sequence reads" from the source DNA by mimicking random variations around an assigned coverage value and with simulated next-generation sequencing lengths and error rates. Because the script simulates variations around a mean-coverage value, genomes with assemblies greater than 20 kbp (or closed genomes) were randomly split in to fragments between 3 kbp and 15 kbp in length using a Python script (split_file.py). For each community, 20 *in silico* metagenomes were created where each genome within the community had a different coverage value. In each iteration of a metagenome for an *in silico* community, organisms were assigned to be either low (randomly assigned a coverage value <10×) or high abundance (randomly assigned a coverage value between 10×–200×) by the script make_config_ini.py. The metagenomes were randomly selected to provide coverage values for the binning tools, with various tests performed on 2–20 *in silico* metagenomes.

Three artificial communities were constructed to test BinSanity and the other tools. The first artificial community selected 50 organisms from distinct species curated from the 60 reference genomes. Further referred to as diverse-mixture-1. In diverse-mixture-1, half of the organisms ($n = 25$) were randomly assigned to be either low or high abundance for each metagenomic sample. Organisms were independently assigned to the low and high abundance categories for each *in silico* sample. A second artificial community with 50 organisms was curated from the 60 reference genomes. This community, henceforth called diverse-mixture-2, assigned all organisms to be low abundance. The last artificial community contained 25 organisms, including four strains of *Escherichia coli* (further

referenced as, strain-mixture). The strain-mixture randomly assigned organisms as low ($n = 13$) or high abundance ($n = 12$).

After the reads for each *in silico* metagenome were generated, the reads were aligned back to the reference genomes using Bowtie2 (*Langmead & Salzberg, 2012*) (v2.2.5; default parameters). The output SAM file was then converted to a BAM file using SAMtools (*Li et al., 2009*) (v1.2 parameters: samtools view -bS file | samtools sort—file). This BAM file was used to calculate the coverage for each contig (reads/bp) via an in-house script (contig-coverage-bam.py) that implements BEDtools (*Quinlan & Hall, 2010*). The determined coverage values were log transformed and results from multiple metagenomes were combined in to a single matrix using an in-house script (cov-combined.py).

Within BinSanity, each contig is evaluated as a possible exemplar based on the coverage. The exemplar is the contig that best represents the contigs clustering with it and can also be referred to as the cluster center. AP is described elsewhere (*Flynn & Moneypenny, 2013*; *Frey & Dueck, 2007*; *Walter, Fischer & Buhmann, 2007*), but in brief, AP takes as input a collection of values where the similarity $s(i, k)$ indicates how well the data point with index $k$ is suited to be the exemplar for data point $i$. The messages sent between points make up either the responsibility $r(i, k)$ or the availability $a(i, k)$ (*Frey & Dueck, 2007*; *Gan & Ng, 2015*). The responsibility is the accumulated evidence that sample $k$ should be the exemplar for sample $i$ (Formula (1)) (*Walter, Fischer & Buhmann, 2007*). The availability (*Walter, Fischer & Buhmann, 2007*) is the accumulated evidence that sample $i$ should choose sample $k$ to be its exemplar, dually considering the evidence of values for other samples that $k$ should be an exemplar (Formula (2)). Two limitations of AP are that it is hard to pinpoint the optimal preference (p) and damping factor. High values of a preference will lead to more exemplars (splitting) and low preferences will lead to a smaller number of exemplars (lumping). When setting a global value for AP, the minimum similarity is typically used as an initial choice for the preference (*Frey & Dueck, 2007*). The damping factor is a number that helps to account for exemplars in periodic variance during the iterative process as well as improves convergence during oscillations (*Mehmood & Bie, 2015*; *Zhengdong & Carreira-Perpinan, 2008*). In addition, AP faces the challenge of time and memory complexity in the order of $O(N^2 T)$ where $N$ is the number of samples and $T$ is this number of iterations until convergence (*Flynn & Moneypenny, 2013*; *Frey & Dueck, 2007*; *Mehmood & Bie, 2015*; *Walter, Fischer & Buhmann, 2007*). This order does not scale for production of a dense similarity matrix.

$$r(i, k) \leftarrow s(i, k) - \max_{k' s.t. k' \neq k} \{a(i, k') + s(i, k')\} \tag{1}$$

$$a(ik) \leftarrow \min \left\{ 0, r(k, k) + \sum_{i' s.t. i' \notin \{i, k\}} \max\{0, r(i', k)\} \right\}. \tag{2}$$

BinSanity consists of two scripts, Binsanity.py and Binsanity-refine.py. BinSanity does an initial clustering of contigs based on the log transformed coverage as produced by contig-coverage-bam.py. First, a Euclidean distance similarity matrix is computed using scikit-learn. This matrix is used as input for AP (accessed via scikit-learn). The resultant

cluster assignments for each contig are then used to generate FASTA files of each set of sequences. Several of the default settings can be modified depending on the nature of the metagenomic samples. Preference (-p) is used to adjust the degree to which AP will group or split contigs with similar coverages. A higher value will lead to a more stringent similarity requirement (i.e., create more clusters), whereas a smaller value will lead to more relaxed similarity requirements (i.e., create less clusters). Testing has shown that a preference value of -10 (-p -10) is successful, if used in conjunction with the refinement script (see below). Maximum iterations (-m) is the total number of iterations performed during clustering, if AP does not identify stable boundaries between clusters. If cluster boundaries are stable for the value given by the convergence iteration parameter (-v), then AP will stop before reaching the maximum iterations. Damping factor (-d) helps to account for contigs oscillating between two cluster centers across multiple iterations. Decreasing the damping factor could lead to uncontrolled oscillation that prevent AP from finding the optimal answer after the maximum iterations is reached.

BinSanity-refine.py is intended to be used following BinSanity.py and incorporates percent GC (%G+C) and tetranucleotide frequencies to re-cluster contigs from high contamination and low completion bins. Convergence iteration, maximum iteration, contig cut off length, and dampening factor parameters are identical to the initial script. The default preference is decreased in this script (-p -25) to account for the extra input data provided by the %G+C and tetranucleotide frequencies. BinSanity-refine.py calculates both %G+C and tetranucleotide frequencies of the provided contigs. The script then proceeds as above.

BinSanity was executed on the log transformed coverage matrix using the script BinSanity.py (-m 4000 -v 400 -d 0.95 -x 1000 -p -10). BinSanity was compared against CONCOCT (*Alneberg et al., 2014*) (v.0.4.1; default parameters), GroopM (*Imelfort et al., 2014*) (v0.3.5; default parameters), MetaBat (*Kang et al., 2015*) (v0.26.3; default parameters), MaxBin (*Wu et al., 2014*) (v2.1.1; default parameters), and MyCC (*Lin & Liao, 2016*). All the methods were used with coverage information, if applicable. BinSanity was tested with and without the refinement script. Initial analysis of the clustering results were conducted via CheckM (*Parks et al., 2015*) for completion, contamination, and strain heterogeneity. For this manuscript, 'contamination' values less than 10% will be referred to as 'redundancy,' as multiple copies of a marker gene may not always represent 'contamination,' but unmeasured diversity in the core genome of a phylogenetic group. After an initial round of binning based on coverage, bins determined by CheckM as highly contaminated or low completion, were subjected to a composition based refinement (BinSanity-refine.py; -m 4000 -v 400 -x 1000 -d 0.95 -p -25). Because reference organisms had known completion and low redundancy estimates, high completion bins were considered to be greater than 90% complete with less than 10% redundancy. Low completion bins were less than 90% complete and less than 5% redundant. Any bins that did not fit in either low completion or high completion were labeled as high contamination. The Binsanity-refine.py script calculates %G+C, tetranucleotide frequencies, and optionally will incorporate coverage to refine high contamination bins and re-cluster low completion bins. Tetranucleotide frequencies

were scaled by 100 and log normalized. Results were evaluated by calculating precision, recall, and V-measure (e.g., harmonic mean) as defined by *Rosenberg & Hirschberg (2007)* using sklearn.metrics.homogeneity_completeness_v_measure (*Pedregosa et al., 2011*) (bin_evaluation.py). Precision measured whether each cluster contains only members of a single class (output = 1, all bins contain only contigs from a single source). Recall measured whether each member of a class is assigned to the same bin (output = 1, only contigs from one source organism are contained in a single bin). The V measure was calculated as the harmonic mean of the precision and recall, allowing for the evaluation of accuracy. An additional measure, the Adjusted Rand Index (ARI) (*Hubert & Arabie, 1985*) was also calculated via sklearn.metrics.adjusted_rand_score (*Pedregosa et al., 2011*) (bin_evaluation.py). The ARI considers similarity between predicted and true cluster labels by creating a contingency table comparing clusters. Within the context of this study, ARI analyzes the four possible situations that can arise when comparing determined cluster labels to the initial reference labels: (1) contigs are assigned to the same group in the reference and in the cluster; (2) contigs are in the same group in the reference and in different groups in the clusters; (3) contigs are in different groups in the reference and are assigned to the same group in the clusters; or, (4) contigs are in different groups in the reference and in different groups in the clusters. This similarity is then adjusted for chance using a probability heuristic. This adjustment accounts for the fact that given random cluster labeling you would expect to get a non-zero ARI. ARI analyzes the relation between elements in each class, in addition to these direct comparison of cluster labels (*Santos & Embrechts, 2009*). The general workflow for affinity-propagation is shown in Fig. 1.

## Infant gut metagenome

BinSanity was tested using samples from a time series study of an infant gut microbiome, previously described by *Sharon et al. (2013)*. Samples were run though BinSanity.py (parameters: -p -10 -m 4000 -v 400 -d 0.95 -x 1000). This same dataset was assessed by *Eren et al. (2015)* and was binned using a human guided strategy via the Anvi'o platform (*Eren et al., 2015*). In an effort to measure the effect of the binning algorithms (and to avoid influencing the results due to the use of different assemblers) the contigs produced by *Eren et al. (2015)* (http://anvio.org/data/) were used as the input for BinSanity (referred to as Eren-contigs). Raw reads were accessed from the NCBI SRA database (SRA052203) and aligned to the Eren-contigs. The coverage matrix was determined as described above. Additionally, the Eren-contigs were binned using CONCOCT, GroopM, MaxBin, MyCC and MetaBat. The *Sharon et al. (2013)* results were retrieved from All genome bins were evaluated via CheckM (*Parks et al., 2015*) and compared to genomes generated by *Sharon et al. (2013)* (http://ggkbase.berkeley.edu/carrol/). The Eren-contigs were Blast searched against the *Sharon et al. (2013)* contigs so that contig ids for each could be visually compared (results available http://merenlab.org/tutorials/infant-gut/). To maintain consistency, the curated bins from *Sharon et al. (2013)* were processed using CheckM, so that all single gene copy based redundancy and completion metrics were consistent.

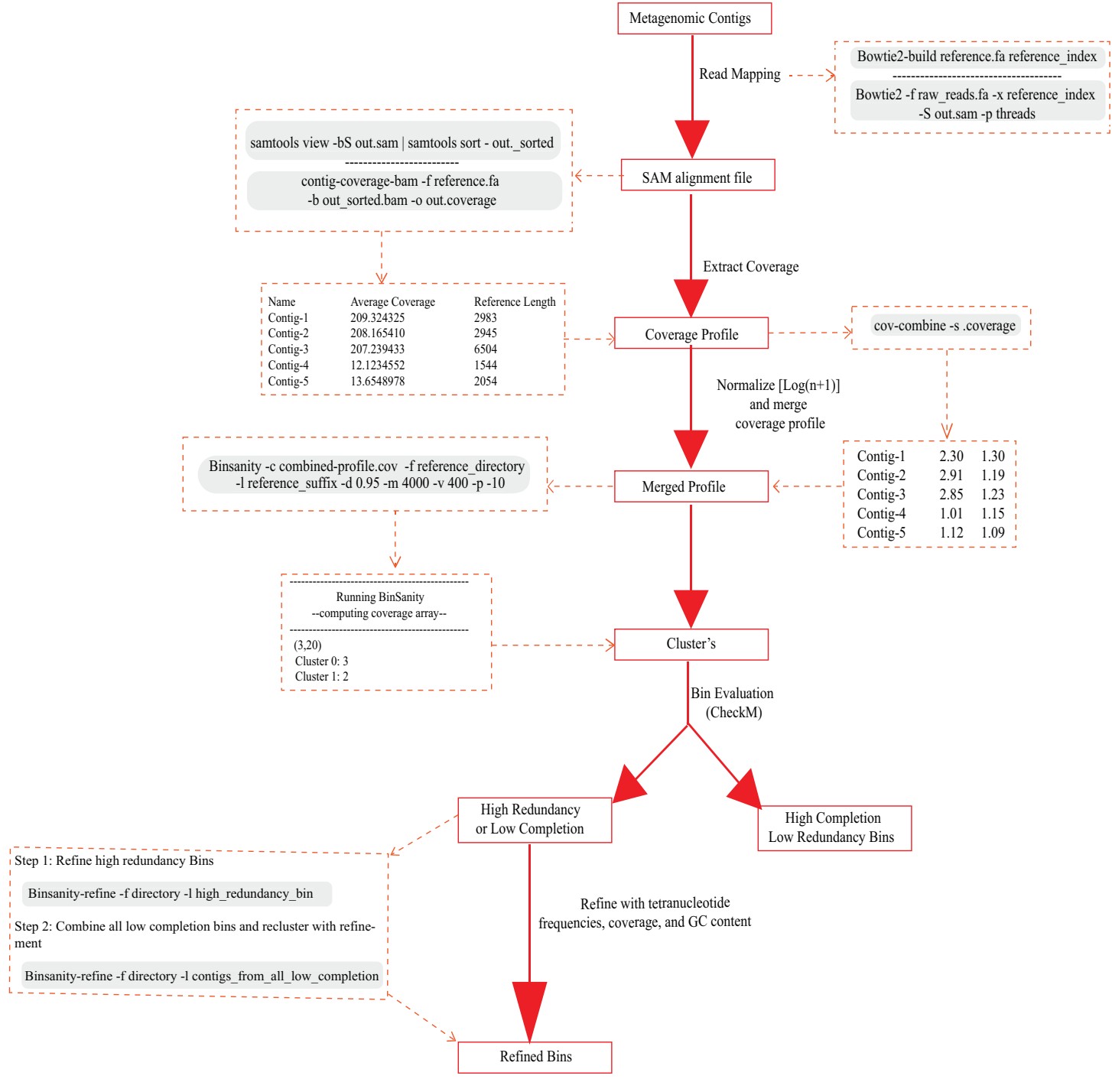

**Figure 1    Workflow for Binsanity indicating all scripts used.**

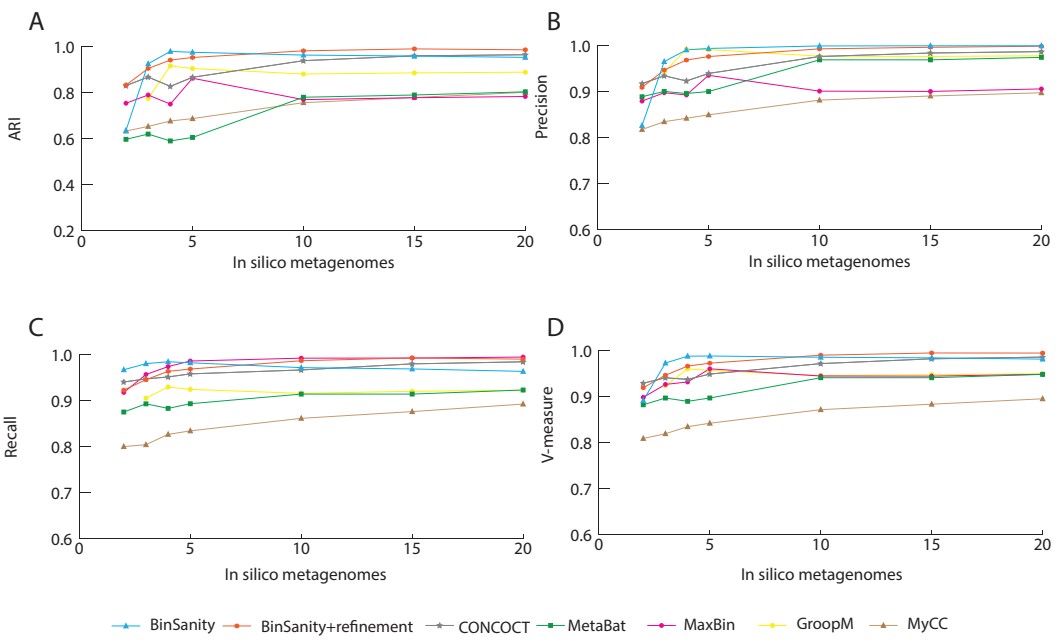

**Figure 2** Stastistical calculations (bin_evaluation.py) showing the adjusted rand index (ARI) (A), precision (B), recall (C), and V-measure (D) for diverse-mixture-1.

# RESULTS AND DISCUSSION

## Species level: diverse-mixture-1

In processing diverse-mixture-1, BinSanity+refinement had near perfect results with an ARI and V-measure of 0.98 using 20 *in silico* metagenomes (Fig. 2). When the number of *in silico* metagenomes was decreased to five, BinSanity had the highest ARI at 0.97, while BinSanity+refinement had an ARI of 0.95. With this number of metagenomic samples, BinSanity produced the highest V-measure score of the binning methods, indicating it most closely reconstructed the reference organisms and had minimal rates of incorrectly assigning contigs. Without the use of refinement, BinSanity produced 54 bins of which one had high contamination (>10%), and five were low completion (<85% complete). When BinSanity+refinement was implemented 52 bins were produced (Table 1). Of those 52 bins, two were less than 20% complete and contained contigs originating from a single reference organism.

In comparison to BinSanity, CONCOCT, GroopM, MyCC and Metabat had high precision and low recall, producing more bins than expected (71–109 bins), while MaxBin had high recall and low precision producing less bins (42), when using coverage data from five *in silico* metagenomes (Table 1).

Of the 47 bins produced by MaxBin, eight were highly chimeric (Fig. 3). When isolating the bins that contained >90% complete genomes, BinSanity produced 46 bins, while MetaBat and GroopM produced 33 and 41, respectively. CONCOCT, overall, had a high accuracy, but had difficulty delimiting closely related species such as *Roseobacter denitrificans* and *R. litoralis*. This difficulty in separating closely related species could be related to the use of a single step clustering protocol, where composition and coverage are

**Table 1 Number of Bins Produced by Each Method for each number of *in silico* metagenomes.**

| *In silico* metagenome | BinSanity | BinSanity+ refinement | CONCOCT | GroopM | MetaBat | MaxBin | MyCC |
|---|---|---|---|---|---|---|---|
| Diverse-mixture-1 (n = 50) | | | | | | | |
| 2 | 32 | 46 | 70 | – | 73 | 44 | 107 |
| 3 | 38 | 51 | 64 | 102 | 74 | 41 | 110 |
| 4 | 52 | 51 | 68 | 109 | 74 | 39 | 106 |
| 5 | 54 | 52 | 71 | 109 | 71 | 42 | 103 |
| 10 | 64 | 53 | 73 | 86 | 81 | 38 | 99 |
| 15 | 51 | 52 | 71 | 87 | 31 | 37 | 98 |
| 20 | 72 | 55 | 69 | 81 | 78 | 38 | 83 |
| Diverse-mixture-2 (n = 50) | | | | | | | |
| 2 | 18 | 46 | 74 | – | 56 | 48 | 104 |
| 3 | 33 | 50 | 70 | 59 | 73 | 44 | 127 |
| 4 | 41 | 50 | 72 | 58 | 71 | 43 | 124 |
| 5 | 46 | 50 | 71 | 92 | 69 | 41 | 123 |
| 10 | 52 | 50 | 62 | 68 | 73 | 43 | 126 |
| 15 | 54 | 51 | 57 | 78 | 74 | 40 | 144 |
| 20 | 55 | 51 | 55 | 60 | 75 | 37 | 160 |
| Strain mixture (n = 25) | | | | | | | |
| 2 | 21 | 17 | 33 | – | 38 | 25 | 85 |
| 3 | 23 | 22 | 34 | 34 | 53 | 18 | 53 |
| 4 | 28 | 25 | 35 | 50 | 46 | 19 | 53 |
| 5 | 30 | 25 | 34 | 63 | 48 | 18 | 55 |
| 10 | 35 | 26 | 32 | 41 | 45 | 22 | 63 |
| 15 | 39 | 26 | 28 | 58 | 47 | 21 | 57 |
| 20 | 42 | 27 | 25 | 41 | 42 | 18 | 46 |

used as equally weighted inputs. Closely related organisms often have similar composition signals, while coverage is reliant on the underlying population of the organisms in the community. This can lead to instances where contigs from similar strains cannot be teased apart using compositional data, but can be separated based on coverage values over multiple samples.

It can be difficult to distinguish strains using coverage based methods if reads are not stringently assigned due to bias within conserved regions and nonspecific alignment. Strict alignment parameters (such as using the—very-sensitive flag in Bowtie2) can be used to prevent false contig assignments and increase fidelity of all the binning methods. Additionally, more coverage information, especially variable coverage data, benefits all the methods, as is evident when analyzing results generated using <5 *in silico* metagenomic samples; all methods decline in accuracy (Fig. 2).

The primary method for generating bins within BinSanity is clustering using coverage values. When the number of *in silico* metagenomes decreases (for example, <5 metagenomes), there is an insufficient amount of information to differentiate between low coverage organisms with similar abundances across multiple samples. At four *in silico*

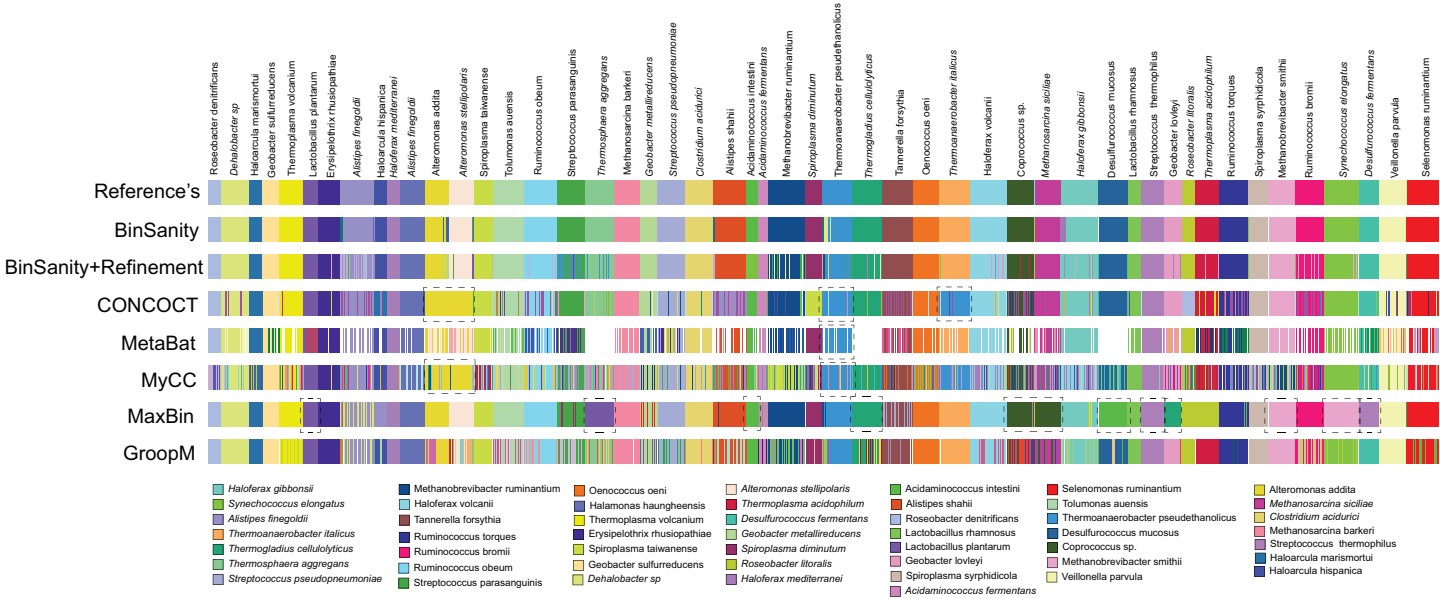

**Figure 3** Clustering results for diverse-mixture-1 BinSanity, BinSanity+refinement, CONCOCT, MetaBat, MyCC, MaxBin, and GroopM at five *in silico* metagenomes (visualized via Anvi'o). Black dashed boxes highlight bins in each method that contained contigs from two or more reference organisms. White represents those contigs that were left un-clustered.

metagenomes, BinSanity grouped organisms with similar coverage profiles together, leading to some bins with high contamination. Utilizing the refinement method to differentiate bins with high contamination increased the precision and recall values when the amount of coverage data was limited. When using refinement with data from two *in silico* metagenomes, BinSanity returned the highest ARI value (Fig. 2).

## Species level: diverse-mixture-2

In diverse-mixture-2 (all organisms <10× coverage), the initial clustering step from BinSanity decreased in accuracy (e.g., decreased ARI, precision, and V-measure) when using data from <10 *in silico* metagenomes, though maintained near perfect ARI scores when ≥10 samples were tested (Fig. 4). This trend was expected, as the convergence of coverage from multiple organisms would lead to contigs from multiple taxa being clustered into the same bin. Utilization of the refinement method, resolved many of these artificial clusters (Fig. S1), such that BinSanity+refinement achieved ARI scores of 0.99 when ≥10 samples were used for clustering and maintained the highest ARI value when 3–5 *in silico* metagenomes were used; at two to three metagenomic samples, BinSanity was outperformed by CONCOCT.

Comparison of CONCOCT, MaxBin, MetaBat, GroopM, BinSanity, and BinSanity+refinement at five *in silico* metagenomes, indicated that BinSanity+refinement produced bins with a higher degree of agreement to the true contig assignments (Fig. 4). At five *in silico* metagenomes, BinSanity (without refinement) produced 46 bins compared to an input of 50 genomes. When refinement was incorporated into the workflow, BinSanity+refinement was able to resolve the 50 bins. BinSanity+refinement

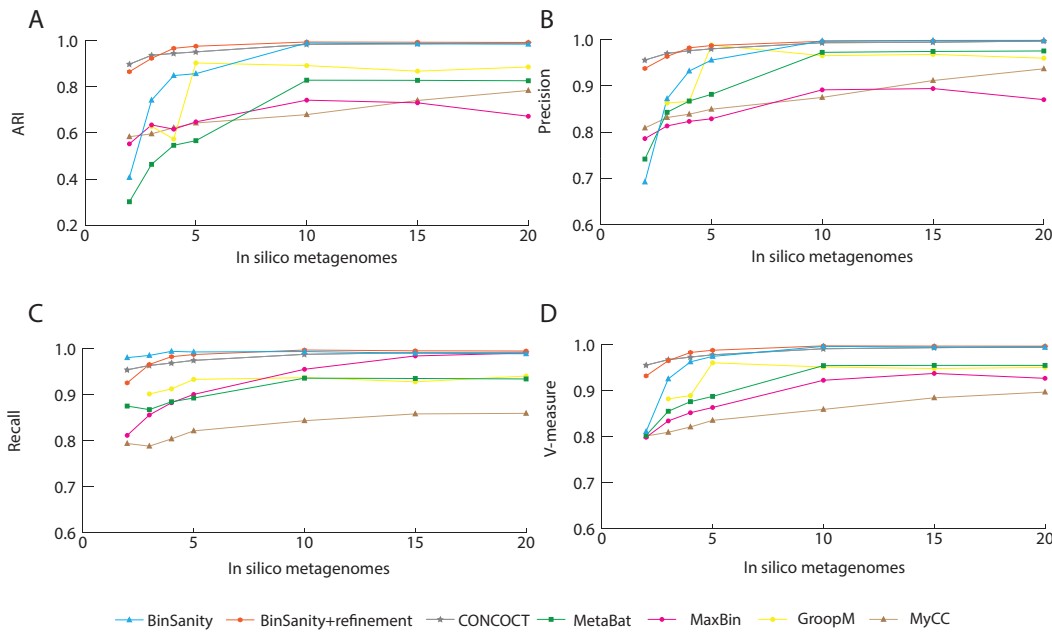

**Figure 4** Statistical calculations (bin_evaluation.py) showing Adjusted Rand Index (A), Precision (B), Recall (C), and V-Measure (D) for diverse-mixture-2.

could accurately split contigs from six organisms that were clustered into two bins during the initial BinSanity step. As with the previous community test, CONCOCT, GroopM, MyCC and MetaBat produced more than the input genomes (69-123 bins), while MaxBin created 41 bins. CONCOCT and GroopM produced results with more accuracy than MaxBin, MyCC, and MetaBat. However, GroopM failed to cluster one organism and over split several other organisms, and CONCOCT clustered two *Desulfurococcus* species and over split several genomes. MetaBat massively over split genomes and had a high percentage of contigs (14.84%) that were not placed in bins. MyCC, similar to MetaBat, over split multiple genomes, but had lower instances of bins containing multiple taxa. These results suggest BinSanity can separate low coverage organisms effectively from a large sample set by conducting a first pass using the standard BinSanity script, followed by refinement of bins with high contamination and/or low completion.

## Strain-level

For the strain-mixture community with 25 organisms (including 4 strains of *Escherichia coli*), BinSanity produced 30 bins when using data from five metagenomes. When refinement was used to re-cluster high contamination and low completion bins, the output was reduced to the target 25 bins. In contrast, CONCOCT, MetaBat, MyCC, and GroopM over split the data (34–63 bins), while MaxBin did not generate the input number of genomes (19 bins). These tools all had lower overall values for the other determined metrics compared to BinSanity+refinement (Fig. 5). BinSanity maintained the highest ARI and V-measure regardless of the number of metagenomes used to generate the coverage values. While GroopM and MetaBat did created more bins than the number of target

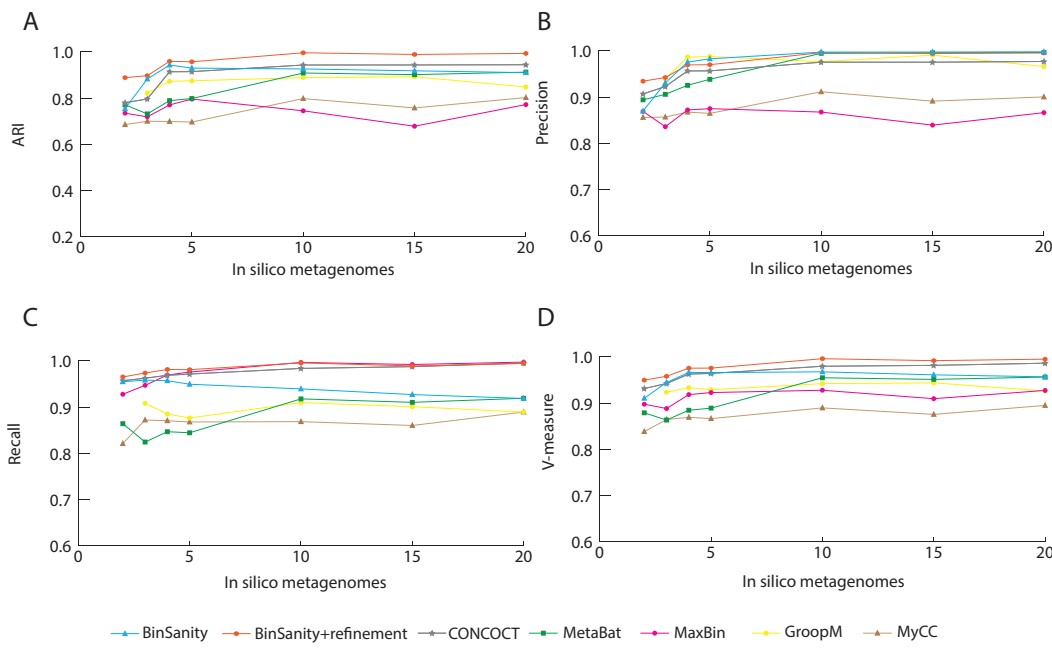

**Figure 5** **Statistical calculations (bin_evaluation.py) for Adjusted Rand Index (A), Precision (B), Recall (C), and V-Measure (D) for the strain-mixture.**

genomes, those bins did have high precision (i.e., a low percentage of bins contained contigs from multiple lineages). MyCC and CONCOCT had difficulty delineating some of the *E. coli* strains and the two *Escherichia* species (Fig. S2).

The primary difficulty for clustering this dataset for all the tested methods was accurately differentiating organisms with strain-level similarity. BinSanity+refinement produced 25 bins. Of the 25 bins produced 3 had high contamination and high strain heterogeneity. One of these bins was 91% complete with 68.39% contamination and 99.50% strain heterogeneity (Table S2). This bin contained contigs from *Escherichia coli 083H1* (4.1%), *E. coli UMN026* (12.3%), and *E. coli 0104 H4* (81%). The second bin was 84.64% complete with 13.79% contamination and 95.83% strain heterogeneity. This bin primarily contained contigs from *E. coli 083H1* (98%) but also contained contigs from *E. fergusonii*. The third bin was 68.97% complete with 8.62 % contamination and 100% strain heterogeneity. This bin contained contigs from *E. coli UMN026* (70%), *E.coli 083H1* (0.84%), and *E. coli 0104 H4* (28%). *E. coli 0104  H4* and *E. coli UMN026* were the least accurately clustered with contigs being placed into two and four bins respectively. MaxBin achieved the best resolution of the *E. coli* strains, but had difficulty clustering other organisms within the community (Fig. S2). Metabat and GroopM had high precision, but an extremely low recall due to high degree of genome splitting. CONCOCT, although approximating the correct results for the other members of the community, largely clustered all 6 *Escherichia* genomes into a single bin.

For the strain-mixture community, GroopM, MetaBat, and MaxBin failed to cluster the most contigs, 261, 56, and 49 contigs, respectively. BinSanity fared better than CONCOCT in accurately representing strains. Based on both the statistics (ARI, precision, and recall)
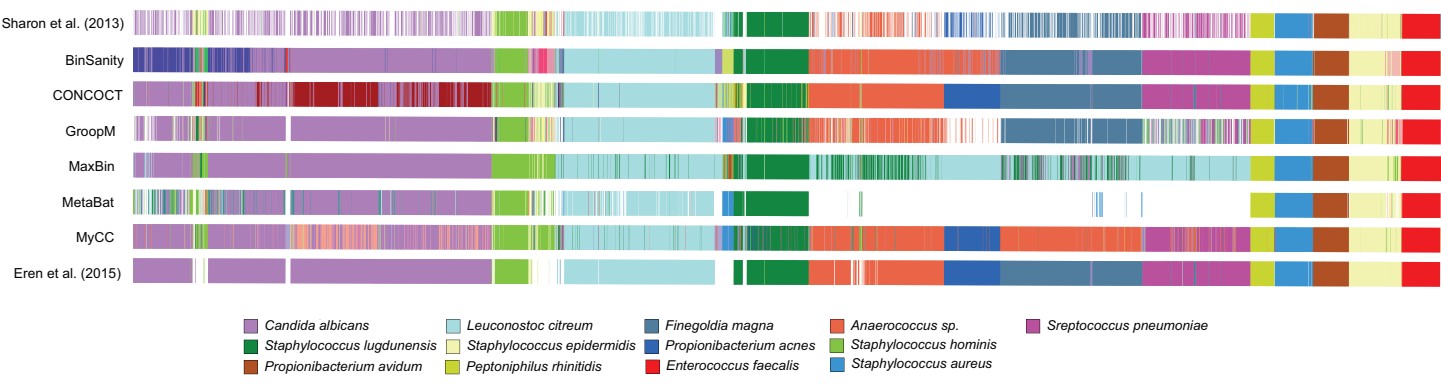

**Figure 6** Clustering of the infant gut metagenome by BinSanity, CONCOCT, GroopM, MaxBin, MetaBat, MyCC, *Eren et al. (2015)* and *Sharon et al. (2013)*. The image was generated through Anvi'o.

and binning output analysis, BinSanity performed better than the current published unsupervised methods for clustering a community with strain-level variation.

## Infant gut metagenome

BinSanity was applied to a metagenomic dataset from a time-series of samples collected from an infant gut environment by *Sharon et al. (2013)* and assembled by *Eren et al. (2015)*. The CLC assembled contigs were processed using BinSanity, CONCOCT, GroopM, MaxBin, MyCC, and MetaBat (Fig. 6). The results from the BinSanity method were additionally compared to the output generated by *Sharon et al. (2013)* and *Eren et al. (2015)* (Table 2). The *Eren et al. (2015)* bins were curated using human guided binning via Anvi'o, and the *Sharon et al. (2013)* genomes were generated via ESOM (*Dick et al., 2009*) and manual curation. Without refinement, BinSanity closely resembled the previously identified bins/genomes.

BinSanity had minor issues in resolving three of bins from the dataset. BinSanity split contigs assigned to *Staphylococcus epidermidis* into two bins, generating a bin with a majority of the *S. epidermidis* contigs that was 89.1% complete, in comparison to 90.3% and 99.8% complete genomes determined via Anvi'o and Sharon et al. respectively. BinSanity without refinement clustered the genome fragments assigned by Sharon et al. as *Propionibacterium acnes* (5.6% complete) and *Anaerococcus* (2.51% complete) into a single bin. These organisms were present at such low abundance, as revealed by their incomplete nature, that BinSanity could not resolve this delineation. When BinSanity+refinement was applied, the results mirrored that of Anvi'o, with a single *Anaerococcus* bin at ~10% complete and *P. ances* bin at 0% complete (refined contig assignments provided in Table S3). *Candida albicans*, a eukaryote, was difficult to cluster accurately for all three methods (Table 2). However, this can be expected as the task of accurately clustering DNA from eukaryotic genomes is currently beyond the scope of BinSanity and many of the methods discussed in this research. BinSanity was able to accurately cluster four bins that mapped back to the *Staphylococcus aureus* virus, *Propionibacterium* virus, *S. epidermidis* virus 013, and *S. epidermidis* virus 014 described by *Sharon et al. (2013)*.

**Table 2  Infant Gut Metagenome CheckM comparison (% completion, % contamination).**

| Bin ID | ESOM (*Sharon et al., 2013*) | BinSanity | Anvi'o (*Eren et al., 2015*) |
|---|---|---|---|
| Staphylococcus aureus_33_1 | 99.51 (0.08) | 95.02 (0.66) | 95.02 (0.66) |
| Staphylococcus lugdunensis_33_1 | 84.10 (0.02) | 58.07 (1.72) | 58.07 (1.72) |
| Staphylococcus epidermidis_32_1 | 99.81 (0.00) | 89.06 (0.00) | 90.28 (2.22) |
| Staphylococcus_hominis_M0480 | 95.39 (0.57) | 97.26 (2.42) | 97.73 (2.19) |
| Peptoniphilus harei_30_1 | 98.95 (0.00) | 100 (0.00) | 100 (0.00) |
| Propionibacterium_63_1 | 97.86 (0.00) | 98.95 (0.00) | 98.95 (0.00) |
| Enterococcus faecalis_37_1 | 99.25 (0.00) | 99.63 (0.00) | 99.53 (0.00) |
| Leuconostoc citreum_37_1 | 45.64 (0.23) | 62.94 (2.57) | 62.80 (2.57) |
| Candida albicans_32_1 | 34.43 (9.48) | 60.89 (26.92) | 61.76 (27.65) |
| Finegoldia magna_32_1 | 29.25 (0.00) | 32.54 (0.29) | 35.43 (0.60) |
| Streptococcus_mitis_38 | 16.45 (0.33) | 25.31 (1.00) | 23.10 (0.25) |
| Propionibacterium_acnes | 5.64 (0.00) | 0 (0.00) | 0.00 (0.00) |
| Anaerococcus_18_1 | 2.51 (0.00) | 11.02 (1.22) | 9.90 (0.00) |
| Archaea_unk | 0.00 | 6.00(0.00) | 0.00 |

BinSanity closely approximated the manually derived Anvi'o results with higher accuracy than the other unsupervised methods. CONCOCT clustered *Anaerococcus* and *Finegoldia magna*, while creating two highly chimeric bins from four other organisms. MetaBat failed to cluster a significant majority of the contigs (69%). MaxBin had difficulty identifying four organisms that were >50% complete and had low contig coverage. GroopM resembled both the BinSanity and Anvi'o results, but overall the bins were less robust and contained less contigs. MyCC had difficulty distinguishing between *F. magna* and *Anaerococcus sp.*

Due to the use of the CLC assembled infant gut contigs generated by *Eren et al. (2015)* and not the original contigs from the *Sharon et al. (2013)* study (contigs are not publically available), some variation in the results the other methods are present. These variations can be seen in the *Staphylococcus* bins. For example, *S. lugdunensis* was determined to be ∼58% complete by BinSanity, Anvi'o, and CONCOCT (MetaBat at 49% complete), but the genome published by Sharon et al. was 84% complete. Overall, BinSanity generated bins reflecting published organisms from this metagenome sampling.

## A note on assigning a preference value & the memory complexity

BinSanity is sensitive to changes in the preference value. The preference value sets limits as to how relaxed or stringent AP should be in deciding the number of cluster centers. Although we found high success using the provided default values for BinSanity, results can be optimized for different sample scenarios by taking in to account the complexity and coverage of the microbial community within a sample. When a high range of coverages exists, the preference can be reduced to prevent over splitting the assemblies. When a low range of coverages exists, the preference can be increased to prevent inaccurate clustering of contigs. If strain-level diversity is high, the preference can be inversely scaled to the number of metagenome replicates (e.g., the more metagenomic samples the lower the preference). Iteratively testing preferences is the best way to find the optimal clustering result while

using BinSanity. Recommendations from the authors of AP suggest a good starting point for the preferences is the median or minimum similarity between the most extreme values (*Frey & Dueck, 2007*). We recommend using BinSanity, with preference values that favor higher recall (e.g., using a lower preference, such as the default value of -10) because the refinement script can the successfully separate organisms with similar coverage profiles.

Due to the implementation of AP, parallel computing options are not currently available. AP is a deterministic algorithm meaning, such that re-running the script on identical data will always yield the same clustering results. Clustering diverse-mixture-1 with 27,643 contigs on a Dell PowerEdge R920 with 1TB of available RAM and Intel Xeon 2.3 GHz processors took 191 min and ~54 GB RAM. However, memory usage increases exponentially with more data points (~100,000 contigs ≈ 1 TB RAM).

## CONCLUDING REMARKS

Experimental testing on both real and artificial communities demonstrated that BinSanity+refinement outperformed the binning methods CONCOCT, MetaBat, MaxBin, MyCC, and GroopM when the coverage values for five or more metagenomic samples are available (In some cases BinSanity outperformed BinSanity+refinement). Below four metagenomes, composition information becomes more essential for BinSanity to correctly assign contigs. With this refinement step, BinSanity can maintain high precision and recall across a variety of community types. Based on the unsupervised binning of the infant gut and artificial communities, BinSanity (and BinSanity + refinement) consistently produces results with higher precision, completeness, and ARI compared to other unsupervised methods. Manually curated results generated similar outcomes, though the time spent manually refining bins can become a limiting factor as microbial community complexity increases. BinSanity had more success at consistently generating accurate genomes from strain- and species-level diversity. The consistency with which BinSanity generates high-quality genomes across varying community structures indicates that it is a good alternative to current methods for clustering of metagenomic data.

### Funding
This work was supported by the Center for Dark Energy Biosphere Investigations (OCE-0939654). The funders had no role in study design, data collection and analysis, decision to publish, or preparation of the manuscript. This is C-DEBI contribution number 360.

### Grant Disclosures
The following grant information was disclosed by the authors:
Center for Dark Energy Biosphere Investigations: OCE-0939654.

### Competing Interests
The authors declare there are no competing interests.

## Author Contributions

- Elaina D. Graham conceived and designed the experiments, performed the experiments, analyzed the data, contributed reagents/materials/analysis tools, wrote the paper, prepared figures and/or tables.
- John F. Heidelberg and Benjamin J. Tully conceived and designed the experiments, contributed reagents/materials/analysis tools, reviewed drafts of the paper.

## Data Availability

GitHub: https://github.com/edgraham/BinSanity/.

## Supplemental Information

Supplemental information for this article can be found online at http://dx.doi.org/10.7717/peerj.3035#supplemental-information.

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
