# Peer review of "BinSanity: unsupervised clustering of environmental microbial assemblies using coverage and affinity propagation"

_PeerJ, doi:10.7717/peerj.3035_

## Round 0.1 · original submission · Major Revisions

Please address the concerns raised by the reviewers, in particular the choice of the preference parameter.

Reviewer 1 ·

Basic reporting

This manuscript describes an algorithm to perform unsupervised clustering of metagenomes using coverages and affinity propagation. The algorithm requires at least two metagenomes and claims to outperform existing tools. My review for this manuscript is as follows.

1. My major concern for this tools is on the preference (-p) parameter that determines how affinity propagation works, as this parameter will surely affect the performance of the algorithm. The authors have contributed a section on describing how to determine this parameter; however it is unclear whether they used the spirit described in this section on their own samples. In other words, are the samples mentioned in the whole manuscript were run by the predicted preference values as described in this section? If, instead, they need to adjust the parameter to "beat" other tools, then I will highly doubt the usability of this tool.

2. Continue from 1. If the preference value needs to be determined beforehand, I do not think the authors can put the word "unsupervised" in their title. This will be at best "semi-supervised." So if the preference value really need user input then please adjust the title to reflect this.

3. Also continue from 1. One way to adjust preference value is to go through some trial runs and determine the best result. This will of course be quite time-consuming, but you alleviate the users from the need to adjust the parameter by themselves. I am not saying that the authors should do this but just throw one possibility for a truly "unsupervised" algorithm.

4. For unknown reason the authors put the algorithms and definition (from line 51 to line 86) in the Introduction part instead of in the Method. For clarify please consider moving these parts into Method.

5. According to Figure 1, genomes with high completion and low contamination are kept while those with low completion and/or high contamination are put into refinement. Perhaps I missed it, but I do not see the actual criteria for determining which bins can be kept or which bins need further processing. Also how about bins with high completion and high contamination?

Experimental design

1. Have the authors compared their results against myCC?

2. What is the run time for the tools?

3. Have the authors tried samples with more than 100 genomes?

Validity of the findings

No comments

Reviewer 2 ·

Basic reporting

In this study, the authors address the topic of binning contigs from metagenomic studies. They contributed towards the development of an unsupervised binning method, which utilizes the clustering algorithm Affinity Propagation (PA) and contigs coverage, using composition data as well to refine bins in studies across several organisms.

Experimental design

The issue addressed here is not novel; methods based on unsupervised binning methods for metagenomics studies are also available. During this year new binning methods have been developed, like the one cited in the article of Lin and Liao, Liu et al [1], Li et al [2], Sun et al [3]. The comparison of BinSanity results with other unsupervised methods ignores all these developments. Authors have decided to compare their results with a set of methods (CONCOCT, GroopM, MetaBat and MaxiBin) that are exactly the same used in previous articles [4][3]. Authors need to further explain why they’ve used the same four methods. The experimental design presented in the article seems well grounded, but it would benefit from a thorough proofreading, being the methodology quite poor, as well.

[1] Liu Y, Hou T, Kang B, Liu F, Unsupervised Binning of Metagenomic Assembled Contigs Using Improved Fuzzy C-Means Method. IEEE/ACM Trans Comput Biol Bioinform. 2016 Jun 7. PMID: 27295684
[Ying Wang, Haiyan Hu, Xiaoman Li, MBMC: An Effective Markov Chain Approach for Binning Metagenomic Reads from Environmental Shotgun Sequencing Projects OMICS. 2016 Aug 1; 20(8): 470–479. Published online 2016 Aug 1. doi: 10.1089/omi.2016.0081]
[3] Lu YY, Chen T, Fuhrman JA, Sun F, COCACOLA: binning metagenomic contigs using sequence COmposition, read CoverAge, CO-alignment and paired-end read LinkAge. Bioinformatics. 2016 Jun 2. pii: btw290.
[4] Lin, H.-H. & Liao, Y.-C. Accurate binning of metagenomic contigs via automated clustering 411 sequences using information of genomic signatures and marker genes. Scientific Reports 6, 412 24175, doi:10.1038/srep24175

Validity of the findings

The authors raise some interesting points that seem worth discussing, but it makes important error in analysis and interpretation. There is important information missing, like explain more in detail the BioSanity script and defining better the different parameters. The current results do not allow the authors to confidently reject other equally viable alternative methods. I would like to see more and deeper examination of the validation of Binsanity as more effective alternative to published unsupervised clustering algorithms.

Additional comments

- Line 48: reference missing.

- In the construction of artificial metagenomes, additional information should be present, like a supplementary table with the list of the 60 reference genomes used to create the in silico microbial communities.

- Line 115 is mentioned 65 reference genomes, that’s correct? before they said about a total of 60 reference genomes.

- Line 117: I found discrepancies between the text in the article and Table S1; in the article it is mentioned diverse-mixture-1 and -2, which, it should correspond to diverse-mixture-2 and -3 in the Table S1, doesn’t it?

- In the Figure 1 the Binsanity script is missing the parameter –x that is cited in the line 128 of the article.

- The script BinSanity–refine.py said that uses G+C content, coverage and tetranucleotide frequencies, are those parameters calculated by the script?

- In the final paragraph, authors conclude that Binsanity should “need to manually optimize a preference by experiment basis”, that should derivative a confusion from that they say BinSanity as unsupervised method.

- Figures 2, 3, 5 and 7: orange and red colours are easily confused.

- Authors need to review the reference list. Sometimes, it is missed the year of publication, like reference 2, 33. References lack of homogeneity, for example in those references from conferences. I suggest that they are written in alphabetic order. They need to pay special attention to spelling, like Table S1.

Overall though, with significant improvements where the authors dig deeper into the results and truly grapple with their implications to suggest a better method for unsupervised binning, this could be an important paper.

---

## Round 0.2 · Minor Revisions

While both reviewers are satisfied with the present version, they both suggest further improvements such as moving some figures to the Supplementary Material and using names more consistent with the names used by checkM. I think that the authors must be consider these suggestions in a new version of the paper, which can then be accepted without further review.

Reviewer 1 ·

Basic reporting

The authors have successfully addressed most of my concerns. I have only one minor question: the metrics that checkM used to evaluate the bins are completeness, contamination, and strain_heterogeneity; however the authors mentioned "completion" and "redundancy" when they are talking about checkM results. I guess the authors mean contamination when they are saying the word redundancy. Please revise the words so that readers who are familiar with checkM do not need to guess the meaning of the word "redundancy."

Experimental design

no comment

Validity of the findings

no comment

Additional comments

no comment

Reviewer 2 ·

Basic reporting

no comment

Experimental design

no comment

Validity of the findings

no comment

Additional comments

The authors have addressed most of the concerns I had with the previous submission, clarifying their novelties in these parts I suggested before. I recommend an accept with the suggestion of moving figures 3,5,7 and 8 to Supplementary material.

---

## Round 0.3 · accepted · Accept

I think that your new method can make a useful contribution to the fast developing field of microbiota analysis.